# Self-Control Moderates the Association Between Perceived Severity of Coronavirus Disease 2019 (COVID-19) and Mental Health Problems Among the Chinese Public

**DOI:** 10.3390/ijerph17134820

**Published:** 2020-07-04

**Authors:** Jian-Bin Li, An Yang, Kai Dou, Rebecca Y. M. Cheung

**Affiliations:** 1Department of Early Childhood Education, Center for Child and Family Science, The Education University of Hong Kong, Hong Kong, China; lijianbin@eduhk.hk (J.-B.L.); rymcheung@eduhk.hk (R.Y.M.C.); 2Department of Applied Psychology, Guangdong University of Foreign Studies, Guangzhou 510006, China; yangan@gdufs.edu.cn; 3Department of Psychology and Research Center of Adolescent Psychology and Behavior, School of Education, Guangzhou University, Guangzhou 510006, China

**Keywords:** risk factor, resilience, cognitive appraisal, self-control, COVID-19, public health concerns

## Abstract

Coronavirus disease 2019 (COVID-19) has caused thousands of deaths in China. Prior research suggests that individuals’ perceived severity of COVID-19 is related to a range of negative emotional and behavioral reactions among the Chinese public. However, scant research has examined the underlying mechanisms. Drawing upon the risk-resilience model, this study proposes that self-control, as a resilient factor, would potentially moderate the association between perceived severity of COVID-19 and mental health problems. Data from a national survey was used to examine this idea. Participants were 4607 citizens from 31 regions in China (M_age_ = 23.71 years, 72.5% female) who completed a national survey at the beginning of February 2020. Results of hierarchical regression showed that after controlling for a number of demographic variables, perceived severity of COVID-19 and self-control were positively and negatively related to mental health problems, respectively. More importantly, self-control moderated the “perceived severity of COVID-19–mental health problems” association, with this link attenuating as the levels of self-control increased. These findings suggest that compared to those with high self-control, individuals with low self-control are more vulnerable and are more in need of psychological aids to maintain mental health in the encounter of the COVID-19 outbreak. Practically, enhancing individuals’ self-control ability might be a promising way to improve individuals’ mental health during the early period of the COVID-19 outbreak.

## 1. Introduction

The outbreak of coronavirus disease 2019 (COVID-19) has not only led to thousands of deaths in China, but it also caused tremendous psychological stress to the Chinese public. In order to medically and mentally combat COVID-19, professional input from various disciplines is needed, including the perspectives from psychological science. A recent study conducted among 4607 Chinese reveals that individuals’ cognitive appraisals, perceived severity of COVID-19 in particular, are related to a number of undesirable emotional (e.g., increase in negative emotion) and behavioral (e.g., increase in sleep problems) reactions [1]. However, little is known about what factors can buffer the negative influence of perceived severity on individuals’ mental health. Studying this issue is important and necessary, as the findings will shed light on at specific factors for intervention programs to target. In this study, drawing upon the risk-resilience model [2], we propose that self-control—one’s ability to override or change his/her inner responses and to interrupt undesired behavioral tendencies [3]—could be a candidate that buffers the negative influence of perceived severity of COVID-19 and mental health problems among the Chinese public. The aim of this study is to examine this idea.

The risk-resilience model proposes that risk and adversity increase the propensity of undesirable outcomes; individuals who have sufficient assets to offset the negative influence of the risk could overturn undesirable outcomes, thus showing resilience [2]. In this model, risk can be defined in diverse ways, including both intrinsic and extrinsic factors such as adverse living condition, the negative life events that have occurred in recent months, massive community trauma, and cumulative risk calculations that combine different kinds of risk factors. The main effect tenet suggests that risk factors are associated with undesirable outcomes such as maladjustment. Nevertheless, the compensatory effect (or moderation effect) tenet assumes that sufficient positive internal and/or external assets could mitigate the burden of an individuals’ life due to risk influences, and thus these individuals would be less affected by the risk and have better outcomes than those who have the same levels of risk but do not possess enough assets. For instance, children reared in risky environments commit more aggressive acts, but assets like positive parenting (e.g., authoritative parenting) buffer the negative influence of risky environment on children’s antisocial behavior [2].

A recent study reveals that individuals’ perceived severity of the COVID-19 outbreak is related to more undesirable emotional and behavioral outcomes in the Chinese public [1], suggesting that perceived severity of COVID-19 may impose a negative main effect on mental health outcomes. This is in line with past studies which found that individuals’ negative appraisals about the incident (e.g., perceived risk, perceived threat, etc.) were related to more mental health problems during the outbreak of severe acute respiratory syndrome (SARS) and Ebola [4,5,6,7]. These studies mainly adopt the main effect tenet to examine the risk and protective factors of outcomes during the encounter of emergent public health concerns. Beyond this, the risk-resilience tenet suggests the necessity and importance to take into account how assets may buffer the negative main effect of risk precursors on health outcomes. This motivated us to examine the potential moderation effect of self-control.

As an asset, self-control ability may serve to regulate the undesirable mental health consequences brought by COVID-19. Previous research suggested that individuals with high self-control have better inhibition and initiatory ability [8], use more positive coping strategies and fewer negative coping strategies [9], and persist more in important life domains [10,11]. In addition, self-control supports the retrieval of restrain standards and deliberative evaluations [12]. Therefore, good self-control is robustly related to a wide range of life outcomes, including better mental health [13]. Besides the main effect of self-control, several aspects of good self-control are particularly relevant to the moderation in the association between risk factors and health outcomes. For instance, planning and forethought could help individuals anticipate and prepare for difficult situations; emotional self-regulation could provide better emotional control in problem situations; the ability to restrain and initiate action is important to deal with the difficulties in problem situations [14]. Prior studies have found that good self-control buffers the influence of negative environment (e.g., peer deviance, negative life events) on undesirable outcomes such as substance use [14,15]. Applying the moderation role of self-control to the case of COVID-19, individuals with good self-control, compared to their low self-control counterparts, are more likely to plan ahead (e.g., preparing for facing the pandemic actively), regulate the negative emotions (e.g., anxiety, fear) induced by the high morbidity and uncertainty at the time when we conducted this study, and strictly adhere to government’s guidelines of prevention and protection (e.g., wearing facial masks, staying at home as much as possible, and maintaining social distancing). These actions may help mitigate the negative influences of perceived risk on individuals’ mental health.

According to the literature reviewed above and the logic of the risk-resilience model, we hypothesized that perceived severity of COVID-19 and self-control would be related to mental health problems in positive and negative directions, respectively. Moreover, we assumed that the negative association between perceived severity of COVID-19 and mental health problems would be less pronounced among those with high self-control compared to their low-self-control counterparts. To examine these hypotheses, we would use data from a recent national survey [1] which examines the Chinese public’s emotional and behavioral outcomes and its antecedents in the encounter of the COVID-19 outbreak. Given that these data were collected from different regions in mainland China, the individual data were nested in regions and therefore we would employ a hierarchical regression analysis to control for the potential influence of the cluster, with a number of demographic variables included in the model as covariates to rule out their influence on the outcome.

## 2. Materials and Methods

### 2.1. Participants and Procedure

We used snowball sampling to recruit participants online to take part in a national survey that investigated Chinese citizens’ emotional and behavioral reactions as well as risk and protective factors during the outbreak of COVID-19 between 2 and 9 February 2020. A total of 4826 Chinese individuals visited our online survey website during the survey period. A total of 219 participants were excluded because they met one of the following criteria: (1) they did not show interest to participate in the study after reading the information sheet; (2) they were not old enough to provide consent form (i.e.,≤16 years); or (3) they showed an obvious responding pattern across multiple consecutive items (e.g., select “3” for all items). Thus, 4607 participants comprised the final sample, 1265 men (27.5%) and 3342 women (72.5%) aged from 17 to 90 years (M_age_ = 23.71 years, SD = 7.29). These participants came from 31 regions in China, with the sample size ranging from 16 (0.3% of the total sample, Ningxia Hui Autonomous Region) to 1386 (30.1% of the total sample, Guangdong Province). Most participants (73.2%) held a bachelor’s degree or above. Regarding physical and mental health condition, 77.3% of participants reported that their current physical health was good or very good. In addition, 94.6% and 99.2% of participants reported that they did not have any history of chronic physical diseases or history of psychiatric/psychological disorders, respectively. Moreover, 97.7% of participants reported they had not been diagnosed with COVID-19 or involved in the pandemic; 2.3% of participants reported they were suspected/diagnosed with COVID-19 or had relatives/friends who were suspected/diagnosed cases.

The ethical committee of the Guangzhou University reviewed and approved this study before data collection (Ethical approval number: GZHU2020001). This study was part of the large survey study which aimed to examine Chinese public’s emotional and behavioral outcomes and its risk and protective factors towards the COVID-19 outbreak [1]. This paper and the large survey study share only one main variable (i.e., perceived severity) and the demographic variables. No other variables overlapped. Thus, we deem that the two papers are substantially distinctive. Over 200 volunteers who majored in psychology in various universities in China helped distribute the survey link on a number of internet platforms, including WeChat, Weibo, QQ, Facebook, forums, etc., after receiving a 3-hour training to get familiar with the research procedure and the contents of the questionnaires. For instance, a volunteer might post the survey link on their WeChat Moment (a function in WeChat which is similar to Facebook and Twitter) so that friends and relatives of that volunteer could participate and further shared the link to their social networks. Participants provided their electronic consent form prior to participation. Voluntary participation was emphasized, and no incentive reward was given to participants. We also stressed anonymity and did not collect any identifiable personal particulars to protect participants’ privacy.

### 2.2. Measures

Mental health problems. The Chinese version of the 12-item General Health Questionnaire (GHQ-12) [16] was used to measure participants’ overall mental health problems over the past 10 days. The scale contains 12 items either in negative or positive wordings. Participants were asked to indicate their situation on these items over the past 10 days compared to their general situation. All items were rated on a four-point scale. According to the scoring system used in Zhang and Wang [16], the first two options of each question indicate that participants feel better than, or as much as, usual. Thus, these two options were coded 0 (i.e., no significant changes in mental health condition). The third and the fourth options indicate participants feel/experience a little and much poorer than usual, respectively, and thus they were coded 1 (i.e., mental health condition slightly worse than usual), and 2 (i.e., mental health condition much worse than usual), respectively. Averaging the items yields a mean score and a higher total score reflects more mental health problems. Sample items are “feeling unhappy and depressed” (negative wording) and “been able to face up to your problems” (positive wording). This measure has been used in prior research showing good internal consistency reliability [16].

Perceived severity. Participants’ perceived severity about COVID-19 was measured with five items developed by our team. Participants indicated their evaluation of how severe they think COVID-19 is in various aspects, including the infection rate, morbidity, mortality, its negative influence on social order, and its negative influence on the economics, on a five-point scale (from 1 = not severe at all to 5 = very much severe). A higher mean score indicates participants perceived COVID-19 to be more severe. Sample items are “how severe you think of the infectious rate of COVID-19 is?” and “how severe you think of the morbidity of COVID-19 is?” This measure has been used in prior research showing good internal consistency reliability [1].

Self-control. The Chinese version of the Brief Self-Control Scale (BSCS) was used to measure participants’ self-control ability [17]. The scale consists of 13 items rated on a five-point scale (from 1 = not like me at all to 5 = like me very much). A higher mean score indicates better self-control ability. Sample items are “I am good at resisting temptation” and “I have a hard time breaking bad habits”. This measure has been used in prior research showing good internal consistency reliability [17].

### 2.3. Data Analysis

We analyzed the data in SPSS 18.0 (IBM, Armonk, NY, USA) and Mplus 7.0 (Muthén & Muthén, Los Angeles, CA, USA) in several steps, with 0.05 as the significance level throughout the analyses. First, we carried out descriptive statistics to capture the levels of each variable. Second, we performed correlation analysis to examine the association between perceived severity, self-control, and mental health problems. Last, given that the individual data were nested in different provinces/regions, we carried out hierarchical regression models to examine the moderation effect of self-control in the association between perceived severity and mental health problems, controlling for a number of demographic variables (i.e., biological sex, age, the history of chronic physical problems, the history of psychiatric/psychological disorders, current physical health condition, educational levels, and relationship with COVID-19). To this end, we first calculated the intraclass correlation and the variance at the within- and between-level. Then, we centered the covariates, independent variable and the moderator using the “groupmean” function [18]. Subsequently, we calculated the product term between the centered independent variable and the centered moderator as the interaction term. Based on this, we fit a full model, with the independent variable, the moderator, the interaction term, and the covariates as level-1 variables. The intercept of the dependent variable was also estimated at level-2 to control for the variance of the dependent variable accounted for by the cluster. Finally, we conducted simple slope tests with 1 standard deviation (SD) below and above the centered mean of the moderator and examined the differences in the magnitude of the slopes (i.e., Slope _1 SD_ vs. Slope _mean_, Slope _1 SD_ vs. Slope _−1 SD_, and Slope _mean_ vs. Slope _−1 SD_) to determine if the simple slopes differ against each other. Since Mplus 7.0 does not accommodate bootstrapping with multi-level models, we used R to calculate the 95% bootstrapping (*N* = 20,000) confidence interval of the simple slopes and their differences. If the confidence interval does not include zero, significant differences between slopes by the levels of the moderator would be tenable.

## 3. Results

### 3.1. Mean Levels of and the Bivariate Correlations Between Mental Health Problems, Perceived Severity, and Self-Control

As shown in Table 1, participants reported low levels of mental health problems (0.19 out of 2) over the past 10 days, relatively high levels of perceived severity of COVID-19 (4.09 out of 5), and medium levels of self-control (3.03 out of 5). Moreover, the results of correlation analyses found that perceived severity of COVID-19 (*r* = 0.19, *p* < 0.001) and self-control (*r* = −0.21, *p* < 0.001) was positively and negatively related to mental health problems, respectively. The effect sizes for these correlation coefficients were small-to-medium, according to Cohen’s (1992) standard [19].

### 3.2. The Moderation of Self-Control in the Association Between Perceived Severity and Mental Health Problems

We conducted hierarchical regression models to examine the association between perceived severity of COVID-19 and mental health problems as well as the moderation of self-control in Mplus. Results of the null model with mental health as an outcome revealed that the intraclass correlation (ICC) of mental health problems was 0.04. The within-level and the between-level model explained 7.1% (*p* < 0.001) and 0.3% (*p* = 0.012) variance of mental health problems, respectively.

The results of the full model which included the independent variable, moderator, dependent variable, and demographic variables are summarized in Table 2. This model was a saturated model. The results showed that after controlling for a number of demographic variables, the main effect of perceived severity on mental health problems was significant yet small (*B* = 0.07, *SE* = 0.01, *p* < 0.001). The main effect of self-control was also significant yet small (*B* = −0.08, *SE* = 0.01, *p* < 0.001). More importantly, the interaction effect between perceived severity and self-control was significant as well, *B* = 0.05, *SE* = 0.01, *p* < 0.001.

Given the significance of the interaction term, we further conducted simple slope tests and the results are displayed in Table 3 and Figure 1. The results showed that the simple slopes were significant when the levels of self-control were low (*B* = 0.10, *SE* = 0.01, *p* < 0.001), medium (*B* = 0.07, *SE* = 0.01, *p* < 0.001), and high (*B* = 0.05, *SE* = 0.01, *p* < 0.001). Furthermore, we compared the differences in the magnitude of these slopes, finding that the slope of perceived severity when self-control was low was significantly larger than the slopes when self-control was at medium and high levels, and that the slope when self-control was at medium levels was also significantly larger than the one when self-control was at high levels. Taken together, the results suggest that perceived severity was linked with only modest mental health issues among the Chinese public. Moreover, the findings also suggest that as individuals’ self-control ability increased, the association between perceived severity and mental health problems decreased. In other words, the negative association between perceived severity of COVID-19 and citizens’ mental health problems was more pronounced among those with low self-control than their counterparts with high self-control.

## 4. Discussion

COVID-19 was declared a Public Health Emergency of International Concern by the World Health Organization at the end of January 2020 [20]. The current study, conducted soon after the declaration, was one of the pioneer surveys around the globe that examined the influence of COVID-19 on individuals’ mental health and well-being. COVID-19 was still largely confined within mainland China when this study was conducted, but it has been spreading worldwide rapidly. At the end of May 2020, there have been already over five million confirmed cases and it has caused more than 350,000 deaths around the world. Now, an increasing volume of studies have examined the influence of COVID-19 on individuals’ mental health and behavior around the world [21,22,23]. However, very few studies have examined the role of perceived risk in mental health and what buffers the influence of perceived risk [1]. In this sense, a unique contribution of this study was the identification of individuals’ perception of risk of COVID-19 as a risk factor of mental health problems and self-control as a buffer of this association.

Prior research found that perceived severity of COVID-19, as a negative appraisal of the event, was associated with more undesirable emotional and behavioral outcomes [1]. This result is also consistent with the findings that negative evaluation of an event, such as high levels of perceived severity, is associated with negative mental health outcomes when individuals encounter a novice virus [5,7,24]. Deepening our understanding of this association, the current study sought to examine self-control as a candidate to moderate the link between perceived severity and mental health problems among the Chinese public. Supporting our hypotheses, the results showed that even after controlling for a range of demographic variables, the association between perceived risk and mental health problems was mitigated by self-control. This finding also supports the main effect and the compensatory effect tenets of the risk-resilience model [2], such that the influence of risk factors (e.g., perceived risk) on individuals’ life outcomes could be buffered by protective factors (e.g., self-control) and thus individuals showed resilience.

We need to point out that the levels of mental health problems found in this study were low. Similarly, research conducted in Italy during the early lockdown period also suggested that the levels of mental health problems (as measured with the Strength and Difficulties Questionnaire) were relatively low [25]. In addition, research conducted in India during the early lockdown also showed that negative sentiments were few and positive sentiments prevailed [26]. Nevertheless, as the lockdown continues and the national economies suffer, individuals’ mental health problems could increase due to various reasons, such as unemployment, reduced salary, and restricted social interaction. Thus, individuals’ mental health problems may change as the situation of COVID-19 develops. In this sense, it is important to continue monitoring individuals’ mental health and offer continuous psychological aid to mitigate the potential negative influence during the post-COVID-19 period.

This study bears both theoretical and clinical implications. Theoretically, the significant interaction suggests that besides the main effect model, the interaction effect model is also important in studying mental health in the context of emergent public health concerns. This is because the effect of a risk factor on health outcomes could be masked or averaged without examining its interaction with other risks and assets, and thereby the likelihood of detecting such an effect is reduced. Clinically, high levels of self-control buffer the association between perceived severity and mental health problems, suggesting that individuals with low levels of self-control are more vulnerable. This implies that those with low levels of self-control would be more in need of psychological aids, which echoes the call by the Chinese government “to provide mental aid to those who need it”. In addition, since self-control is a crucial asset, enhancing individuals’ self-control ability might help buffer the influence of negative appraisal and improve the public’s mental health. We need to point out that although self-control has its heritable underpinnings, self-control can still be shaped by the environment [27] and develops within individuals throughout the life span [28]. Prior research found some programs to be useful in improving individuals’ self-control effectively, such as mediation [29] and comprehensive self-control training [30]. In addition, this study also identified the populations who are vulnerable to mental health problems during the outbreak of COVID-19. These people are females, the elderly, individuals with history of physical, psychiatric, and psychological disorder, and those involved in COVID-19 (e.g., those diagnosed with COVID-19 and relatives/friends of confirmed cases of COVID-19). Hence, these vulnerable populations, especially those with low levels of self-control, are in higher need of receiving psychological aids. When practitioners develop and implement the intervention and prevention programs, they should be highly flexible in terms of modes, materials, and schedules. This is because experts have thought that COVID-19 will last for a substantial period of time [31], which implies that there will be a regular anti-COVID-19 period and activities such as intervention and prevention programs are expected to cater for this special circumstance.

We must acknowledge that the current study has several limitations. First, the self-report data increases the common method bias and the cross-sectional nature prevents us from inferring causality. Therefore, future research should employ multiple informant methods and longitudinal design to further strengthen the findings. We need to stress again the need to continue monitoring individuals’ mental health because of social changes induced by various reasons (e.g., the emergence of COVID-19) can be closely linked with human development [32]. Hence, it would be promising if future research investigates the dynamic relationship between the development of COVID-19 and individuals’ mental health as well as other developmental outcomes. Second, we did not employ a probability-sampling technique to recruit participants and sample sizes from different provinces/regions vary greatly, which may cause selection effect and affect the estimation of the between-level variance. In this sense, readers should interpret the findings with caution and a more balanced sample size from different provinces/regions in future research is desirable. Third, some populations were over-representative, such as females and university students, which may cause some bias in the results. For instance, educational levels are found to be related to more perceived risks and more compliance for the preventive behavior [1], and therefore including a more educated sample may affect the results to some extent. Although we statistically controlled for the potential influence of demographic variables, a more sophisticated design and a more representative sample would be desirable to achieve even more robust findings. Nevertheless, the current findings provide important early evidence about the mental health problems in the Chinese public and how these problems are jointly explained by both risk (i.e., perceived severity) and asset (i.e., self-control) during the outbreak of COVID-19. As the virus currently is spreading worldwide, the current findings could be of potential use for further research around the globe.

## 5. Conclusions

In conclusion, perceived severity of COVID-19 relates to more mental health problems and high levels of self-control buffer this association. We believe that these findings bear important implications in understanding and improving the public’s mental health during the encounter of emergent public health concerns.

## Figures and Tables

**Figure 1 ijerph-17-04820-f001:**
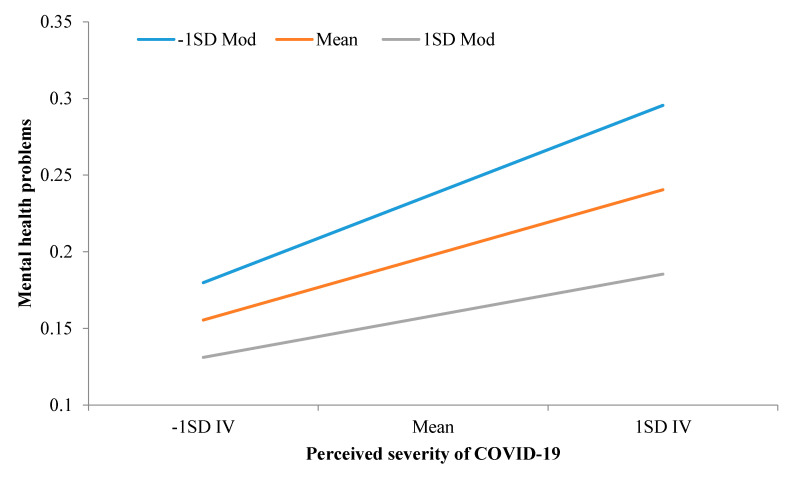
The association between perceived severity of coronavirus disease 2019 (COVID-19) and meatal health problems by self-control. Note. IV = perceived severity of COVID-19; Mod = moderator (i.e., self-control).

**Table 1 ijerph-17-04820-t001:** Descriptive statistics and bivariate correlations of mental health problems, perceived severity, self-control, and demographic variables.

Variables	1	2	3	4	5	6	7	8	9	10
1. Sex										
2. Age	−0.08 ***									
3. Phy. history	0.06 ***	−0.10 ***								
4. Psy. history	−0.01	0.01	0.08 ***							
5. Health con.	−0.04 **	−0.03	0.18 ***	0.08 ***						
6. Education	−0.00	−0.13 **	−0.03 *	0.03 *	0.02					
7. Rel. w. COVID-19	0.01	0.09 **	−0.02	−0.03 *	−0.03 *	−0.01				
8. Mental health problems	0.07 ***	0.04 *	−0.08 ***	−0.09 ***	−0.19 ***	0.02	0.03 *			
9. Perceived severity	0.11 ***	0.09 ***	0.00	−0.00	−0.04 **	−0.03 *	−0.00	0.19 ***		
10. Self-control	−0.01	0.21 ***	0.05 **	0.06 ***	0.21 ***	−0.06 ***	0.03	−0.21 ***	−0.10 ***	
Cronbach’s α	-	-	-	-	-	-	-	0.86	0.84	0.84
Min.	1.00	17	1.00	1.00	1.00	1.00	1.00	0.00	1.00	1.31
Max.	2.00	90	2.00	2.00	5.00	6.00	2.00	2.00	5.00	5.00
M	-	23.71	-	-	-	-	-	0.19	4.09	3.03
SD	-	7.29	-	-	-	-	-	0.27	0.59	0.50

Note. Sex: 1 = male, 2 = female; Phy. history = history of chronic physical diseases: 1 = yes, 2 = no; Psy. histroy = history of psychiatric/psychological disorder: 1 = yes, 2 = no; Health con. = current physical health condition, from 1 = very poor to 5 = very good; Education: 1 = junior middle school and below, 2 = high school degree, 3 = college degree, 4 = bachelor’s degree, 5 = master’s degree, and 6 = doctoral degree. Rel. w. COVID-19 = relationship with COVID-19: 1 = not relevant/infected, 2 = suspected/diagnosed cases or had relatives/friends who were suspected/diagnosed cases. * *p* < 0.05; ** *p* < 0.01; *** *p* < 0.001.

**Table 2 ijerph-17-04820-t002:** Hierarchical regression model of the association between perceived severity and mental health problems and the moderation effect of self-control.

Predictors	*B*	*SE*	*p*
Sex	0.04	0.01	<0.001
Age	0.00	0.00	0.048
Phy. history	−0.04	0.02	0.025
Psy. history	−0.17	0.07	0.013
Health con.	−0.04	0.01	<0.001
Education	0.01	0.01	0.062
Rel. w. COVID-19	0.08	0.03	0.013
Perceived severity	0.07	0.01	<0.001
Self-control	−0.08	0.01	<0.001
Perceived severity × self-control	−0.05	0.01	<0.001

Note. Sex: 1 = male, 2 = female; Phy. history = history of chronic physical diseases: 1 = yes, 2 = no; Psy. histroy = history of psychiatric/psychological disorder: 1 = yes, 2 = no; Health con. = current physical health condition, from 1 = very poor to 5 = very good; Education: 1 = junior middle school and below, 2 = high school degree, 3 = college degree, 4 = bachelor’s degree, 5 = master’s degree, and 6 = doctoral degree. Rel. w. COVID-19 = relationship with COVID-19: 1 = not relevant/infected, 2 = suspected/diagnosed cases or had relatives/friends who were suspected/diagnosed cases.

**Table 3 ijerph-17-04820-t003:** Summary of simple slope tests by the levels of self-control and the difference in simple slopes.

Simple Slope Tests and Comparison	*B*	*SE*	*p*	95% Bootstrapping CI (*N* = 20,000) ^b^
Simple slopes				
Low self-control (−1SD)	0.10	0.01	<0.001	[0.080, 0.116]
Medium self-control (mean) ^a^	0.07	0.01	<0.001	[0.058, 0.086]
High self-control (1SD)	0.05	0.01	<0.001	[0.028, 0.064]
Comparison between simple slopes				
Difference between low and medium self-control	0.03	0.01	<0.001	[0.015, 0.037]
Difference between low and high self-control	0.05	0.01	<0.001	[0.029, 0.075]
Difference between medium and high self-control	0.04	0.01	<0.001	[0.015, 0.037]

Note. ^a^: medium self-control refers to the centered mean. ^b^: 95% CIs were calculated with R.

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
