# Peer review of "Self-Control Moderates the Association Between Perceived Severity of Coronavirus Disease 2019 (COVID-19) and Mental Health Problems Among the Chinese Public"

_ijerph, 2020, doi:10.3390/ijerph17134820_

Round 1
Reviewer 1 Report
The research team aimed to test an interesting hypothesis if high self-control could modify mental health problems as a result of perceived severity of COVID-19. The study was appropriately designed, with the discussion and conclusion adequately presented to highlight potential use of the research findings. In order to further improve the quality of this manuscript, the authors may want to clarify the following:
- Participants:
- The authors mentioned that the sample is not representative of the Chinese population. Please comment on the overall selection of sampling frames used in the recruitment process – there is a risk of selection bias if it was through the network of the volunteers who majored in psychology.
- Since no identifiable personal information was collected, how would the research team ensure that a survey is not completed by a same participant more than once?
- Measurement tools: The authors reported the Cronbach’s α of the 3 key measures as the value in this study in the methods section. Please explain why internal consistency of the tools (questionnaires) were not tested prior to using them in the study. Results from the study should be reported in the results section and not the methods section.
- Results:
- Please report the overall fit of the model. What is the adjusted R-squared value?
- When there is a significant interaction term, please do not elaborate on the main effects alone because their effects are modified by the other covariate. Please delete the extended interpretation.
- Please also refrain from exaggerating the significant yet small effect (the B coefficients are less than a point of the mental health outcome). The authors could translate the statistical significant results (a very small change in the mental health score) in clinical terms, e.g. the potential impact on the prevalence of severe mental health issues amongst Chinese.
Reviewer 2 Report
I would like to congratulate the authors on a very well-written article and presentation.
The article is of immense importance in the times we live in. It is a current and valuable study and being one of the initial studies on mental health globally. Working in the field of psychology, this study is well prepared to make an impact in the mental health field. The title is clear and refers to the content of the article. The abstract is a concise summary of the article, methodology, findings and discussion. It is my opinion that the technical standard of the article is high and I could find very little fault with the methodology, findings and discussion.
The strong points of the article lie in the thorough literature review which included several previous studies with a fair mix of new and older sources. The methodology is clearly set out and the sample size of 4706 participants, is sufficiently large to draw valid conclusions from the findings.
Although the results are sound and clearly explained, it is suggested that the authors mention which version of SPSS was used, ie, 22 or 25, etc.
The authors highlight the limitations of the study in that measures used self-reporting data and might have implications for bias an state that future research should include multiple tools or measures.
The discussion made for very interesting reading. Please check line 263, as I suspect "inconsistent" is meant not " in consistent".
Reviewer 3 Report
Publication is prepared correctly, complies with the requirements and is important in today's social context of the psychologically infected society by COVID-19 and is recommended to publish.
Reviewer 4 Report
This paper is interesting. I have only two small comments. In the descriptive statistics, it is not appropriate to report the mean and standard deviation for categorical variables. Secondly, I would have liked to have seen some attention in the discussion to the whether or not education links to self control. The sample was very skewed in terms of educational background. Has education been linked to self control before? Are more educated people more or less likely to be aware of the risks and to adhere or not adhere to preventative practices?
